# Copper supplementation enhances pigmentation and induces dopamine production in ARPE19

Hironori Uehara ⬤*, Baila Shakaib, Sangeetha Ravi Kumar, Bonnie Archer, Balamurali Ambati

Phil and Penny Knight Campus for Accelerating Scientific Impact, University of Oregon, Eugene, Oregon, United States of America

* huehara@uoregon.edu

## Abstract

Non-neuronal dopamine production has not been understood despite dopamine function in non-neuronal tissues. Tyrosinase is a non-neuronal enzyme which converts tyrosine to L-DOPA (l-3,4-dihydroxyphenylalanine) and L-DOPA to l-dopaquinone for further melanin production. Since L-DOPA is a dopamine precursor in neurons, we hypothesized that tyrosinase-derived L-DOPA could alternatively be converted to dopamine. Therefore, this study investigated whether copper supplementation enhanced pigmentation and induced dopamine production via tyrosinase activation in APRE19 cells. Copper is known as a tyrosinase cofactor. In two separate experiments, we cultured ARPE19 in 1% FBS/DMEM with/without 10 µM copper sulfate for approximately 100 days. After 40–50 days, slight pigmentation with copper treatment was confirmed in the cell pellets, while no pigmentation was observed in the non-copper control. After 90–100 days, the pigmentation in the copper treatment group was obvious, while minimal pigmentation was observed in the non-copper control. Dopamine was not detected at 40–50 days in either group, while it was detected after 90–100 days of culture only in the copper-treated group. Tyrosinase mRNA expression was confirmed in both groups at a similar level, while tyrosinase protein expression was significantly higher in the copper treatment group than in the non-copper control. Thus, we determined that copper supplementation efficiently enhances pigmentation and induces dopamine production in long-term culture ARPE19, likely due to increased tyrosinase protein expression and activity. This is the first report showing the significance of copper in non-neuronal dopamine production of RPE cells, which suggests that tyrosinase may be responsible for non-neuronal dopamine production.

## Introduction

Dopamine is an indispensable neurotransmitter in the central nervous system for motivation and learning, in which dopamine behaves as a reward mediator [1,2]. Various disorders, including Parkinson's disease, drug addiction, and attention deficit

**Data availability statement:** All relevant data are within the manuscript.

**Funding:** Supported by Knight Campus startup funding and in part by the NIH/NEI (R21EY034967).

**Competing interests:** The authors have declared that no competing interests exist.

hyperactivity disorder, are associated with dopamine [3–6]. In addition to the central nervous system, dopamine is functional in peripheral tissues as a vasostimulant and immune modulator [7–11]. In fact, various dopamine receptors are expressed in peripheral organs [12]. However, peripheral dopaminergic neurons have not been clearly identified. Also, due to the instability of dopamine [13,14], it is unlikely that dopamine in the brain transduces into peripheral tissues, suggesting the presence of non-neuronal dopamine biosynthesis. One potential pathway is dopa decarboxylase (DDC, also known as aromatic L-amino acid decarboxylase (AADC)), the enzyme converting l-3,4-dihydroxyphenylalanine (L-DOPA, also known as Levodopa) to dopamine, which is expressed in various tissues [15–20]. Nevertheless, the origin of L-DOPA in peripheral tissues is unclear.

In the retina, a subpopulation of amacrine cells, which localize in the inner plexiform layer, are dopaminergic neurons [21]. Amacrine-derived dopamine can be a neurotransmitter and support other retinal neurons [22,23]. Prior animal studies demonstrate that the dopamine pathway can be associated with eye growth [24–26] and choroidal thickness [27,28], which are related to myopia development [29,30]. In humans, GWAS (genome-wide associated study) revealed that dopamine receptor D1 (DRD1) is associated with refractive error [31]. Hence, while dopamine signaling likely has a significant role in myopia development [32], it is uncertain whether amacrine-derived dopamine can reach the choroid traversing the retinal pigmented epithelium (RPE) and Bruch's membrane and control entire eye growth.

In this study, we hypothesized that tyrosinase in RPE could mediate dopamine biosynthesis. In neurons, the dopamine biosynthesis pathway is well established [33]. Tyrosine hydroxylase converts tyrosine to L-DOPA and, subsequently, L-DOPA to dopamine by DDC (Fig 1, top). In non-neuronal systems, cells in organs such as the kidney, gastrointestinal tract, and immune system can also produce dopamine [34–36]. This dopamine acts locally to regulate ion transport, oxidative stress, and immune responses; however, the exact mechanisms of dopamine synthesis outside the nervous system are still unclear and may differ between tissues [36–38]. As a non-neuronal dopamine biosynthesis pathway, we hypothesized that tyrosinase could be involved in dopamine biosynthesis in RPE (Fig 1, bottom). Tyrosinase is a key enzyme for melanin production [39,40]. Tyrosinase converts tyrosine to L-DOPA and subsequently L-DOPA to L-dopaquinone for further melanin synthesis. We thus hypothesized that tyrosinase-derived L-DOPA could alternatively be converted to dopamine. Since copper is a cofactor for tyrosinase, supplementing copper in the culture medium can drive tyrosinase activity. ARPE19 was used in this study to test the above hypothesis. The ARPE19 cell line, established from a 19-year-old human donor, is commonly used in retinal research due to its ability to mimic several features of native RPE cells [41]. Although not identical to primary RPE cells, including induced pluripotent stem cell (iPSC)-derived RPE cells, ARPE19 remains a useful in vitro model for investigating retinal disorders, inflammation, and oxidative stress [42]. Also, ARPE19 is known to mature in long-term culture and express tyrosinase after at least 14 weeks [43,44]. Therefore, we used a long-term culture of ARPE19 to examine whether copper supplementation enhances pigmentation and induces dopamine synthesis.

**Fig 1. Canonical dopamine biosynthesis pathway in neurons and our hypothesized non-canonical pathway in RPE.**

## Materials and methods

### Cell culture and observation for pigmentation

ARPE19 was obtained from ATCC (Manassas, VA. Cat.no: CRL-2302) and cultured following the manufacturer's instructions. For the long-term culture, we used 1% FBS (Cytiva Life Sciences, Marlborough, MA. Cat.no: SH30396.03)/DMEM (Cytiva Life Sciences. Cat.no: SH30243.01) with Antibiotic-Antimycotic (anti-anti, Thermo Fisher Scientific, Waltham, MA. Cat.no: 15240–062). $0.1 \times 10^6$ or $0.15 \times 10^6$ cells were plated in 6-well plates with 2mL of culture medium. After one day, the culture medium was changed to 1%FBS/DMEM + anti-anti with/without 10 μM copper sulfate (Sigma-Aldrich, Inc. St. Louis, MO. Cat.no: 209198). Copper lactate was obtained from City Chemical LLC (West Haven, CT. Cat.no: C4738). The medium was changed every 3–4 days, typically Monday and Friday, until the cells were harvested. Pigmentation was observed by our eyes and bright-field microscopy (EVOS Cell Imaging System, Thermo Fisher Scientific).

### Copper toxicity assay

CCK-8 and crystal violet assays were conducted to evaluate copper toxicity. For the CCK-8 assay (Dojindo, Tokyo, Japan. Cat no: CK04), 10,000 cells of ARPE19 were plated in 96-well plates using 100 μL of 10%FBS/DMEM/F12 with/without 10μM copper sulfate. Since a previous study indicated copper inhibits WST-8 (water-soluble tetrazolium salt) reduction [45], before the CCK-8 assay, we washed the cells twice with 10%FBS/DMEM/F12 and applied 100 μL of culture medium without copper sulfate. Then, 10 μL of WST-8 was applied. After two hours of incubation in a $CO_2$ incubator at 37°C, OD450 was measured using a plate reader.

For crystal violet assay, 50,000 ARPE19 cells were plated in 24-well plates with 10%FBS/DMEM/F12. After 1 day of incubation, the medium was changed to 1%FBS/DMEM with/without 10μM copper sulfate. The cells were fixed with 4%PFA at the indicated time. After washing the cells with water twice, the cells were stained with 1% crystal violet (Thermo Fisher Scientific, Cat.no: C581-25)/25% methanol/water for 25 minutes at room temperature. After washing

 

the plates with water four times, the plates were dried. 500 µL of 1% acetic acid was used for crystal violet elution. OD590 in 100 µL of the elution was measured using a plate reader.

### Dopamine enzyme-linked immunosorbent assay (ELISA)

Dopamine high-sensitive ELISA was obtained from Immusmol (Bordeaux, France. Cat.no: BA E-5300R). 800 µL of the conditioned medium from the 6-well plate was mixed immediately with 3.2 µL of 1 M sodium metabisulfite (Sigma-Aldrich, Inc. Cat.no: S9000) and 1.6 µL of 0.5 M EDTA (Sigma-Aldrich, Inc. Cat.no: 15575−038) to prevent dopamine degradation. The dopamine assay was started within 30 minutes after harvesting the culture medium. 750 µL was used for dopamine assay according to the manufacturer's instructions.

### Reverse transcription polymerase chain reaction (RT-PCR)

Total RNA was purified using Direct-zol RNA Miniprep kits with DNase treatment (Zymo Research, Irvine, CA. Cat.no: R2052). cDNA (complementary DNA) was synthesized using iScript™ cDNA Synthesis Kit (Bio-Rad Laboratories, Inc. Hercules, CA. Cat.no: 1708891). Standard *Taq* DNA Polymerase (New England Biolabs, Ipswich, MA. Cat.no: M0273L) was used for PCR. Each primer sequence is Human Tyrosinase F: 5'-ACCCATTGGACATAACCGGG-3', R: 5'-TCTTGAAAAGAGTCTGGGTCTGA-3'. Human DDC F: 5'- CCCTGGAGAGAGACAAAGCG-3', R: 5'- CGGAACTCAGGGCAGATGAA-3'. Human DBH F: 5'- GAGACCGCCTTCATCCTCAC-3', R: 5'-ACATGCGGATCTCCTGGAAG-3'. Human TH F: 5'- GTGTTCCAGTGCACCCAGTA −3', R: 5'- TTACACAGCCCGAACTCCAC −3'. Human GAPDH F: 5'- CAGCCTCAAGATCATCAGCA-3', R: 5'- TGTGGTCATGAGTCCTTCCA-3'. Tyrosinase and GAPDH were amplified with 30 cycles, while the PCR cycle number for the other genes was 40 cycles.

### Western blot and melanin quantification

ARPE19 in a 6-well plate was harvested using 250 µL RIPA buffer (Sigma-Aldrich, Inc. Cat.no: R0278). Protein concentration was determined by Pierce™ BCA Protein Assay kits (Thermo Fisher Scientific. Cat.no: 23225). Equal amounts of protein (15 µg) were run in Bolt™ 4–12% Bis-Tris Plus Mini Protein Gels (Thermo Fisher Scientific. Cat.no: NW04120) using Mini Gel Tank and Blot Module Set (Thermo Fisher Scientific. Cat.no: NW2000), and the proteins were transferred to PVDF membrane following the manufacture instruction. After blocking with 5% nonfat dry milk/TBS, we used 1:200 mouse anti-tyrosinase antibody (Santa Cruz Biotechnology, Inc. Dallas, TX. Cat.no: sc-20035) for tyrosinase and 1:3000 rabbit anti-GAPDH antibody (Abcam, Eugene, OR. Cat.no: ab9485) for GAPDH internal control. After staining with an appropriate secondary antibody conjugated with horseradish peroxidase, the Azure 280 Imaging System (Azure Biosystems, Dublin, CA) was used for image capture with SuperSignal™ West Pico PLUS Chemiluminescent Substrate (Thermo Fisher Scientific. Cat.no: 34577).

Since melanin in cell culture is often soluble, we directly measured the OD500 in the cell lysate/RIPA buffer [46]. The protein concentration was adjusted at 2 mg/mL. For the standard, the melanin (Sigma-Aldrich, M8631) was dissolved in 1N NaOH at 10 mg/mL and then diluted to 100 µg/mL, 25 µg/mL, and 6.25 µg/mL. Our standard and spike recovery test in the cell lysate found that 6.25 µg/mL was the detection limit. 80 µL of cell lysates and standards were plated in a 96-well plate, and OD500 was measured with a plate reader.

### Statistical analysis

All results obtained were analyzed using Microsoft Excel. The results in the figures were presented as averages with standard deviation. Student's t-test was used for comparison of averages accompanied with analysis of variance (ANOVA) for multiple group comparisons. $p < 0.05$ was considered statistically significant.

## Results

### Copper supplementation enhanced pigmentation in ARPE19

ARPE19 is known to mature in 1% FBS/DMEM over a long-term culture [43,44]. Importantly, the short-term culture of ARPE19 did not show tyrosinase expression, while a long-term culture of ARPE19 expressed tyrosinase. Therefore, we cultured the ARPE19 cells with/without 10 µM copper sulfate for approximately 100 days. Copper toxicity in the short term (1–14 days) was not observed by CCK-8 (Cell Counting Kit-8) assay (S1 Fig) and crystal violet assay (S2 Fig). Also, the protein amounts in the cell lysis did not decrease after the long-term (105 days) copper treatment, which indicated copper did not show significant cell toxicity in our condition (S3 Fig). Fig 2A shows 40-day-cultured cell pellets. Although it was hard to confirm the color change under the microscope, the cell pellets with copper sulfate were darker than those without copper sulfate. At 65 days, some pigmentation could be confirmed with our naked eye (Fig 2B) and bright field microscopy (Fig 2C). At 90 days, strong pigmentations were observed in copper-supplemented cultures (Figs 2D, 2E, and 3), while only slight pigmentation was confirmed in non-copper cultures. Furthermore, melanin concentration was quantified (Fig 4). Long-term copper treatment significantly increased melanin production. Thus, the copper supplement efficiently enhanced the pigmentation of ARPE19 cells, which indicated that copper enhanced tyrosinase activity.

### Copper supplement induces dopamine production in ARPE19

Next, we examined whether copper supplements induce dopamine production using a dopamine ELISA. Long-term copper treatment over 90 days induces dopamine production in ARPE19 (Fig 5A), while non-copper samples showed less dopamine than the lower limit of analytical sensitivity (3.3 pg/mL, according to the manufacturer). We also tested a 45-day culture medium, but the dopamine level was below the detection limit. Lastly, after 104 days of cell culture without copper, we added 10µM copper sulfate or copper lactate and tested dopamine concentration (Fig 5B). We found that the short-term addition (4 days) of copper, even if after long-term culture, did not induce dopamine synthesis. This result indicated that long-term copper supplementation is a critical factor for dopamine production.

### Copper treatment promotes tyrosinase protein expression but not mRNA expression

To understand how copper supplement induces dopamine biosynthesis in ARPE19, we examined several key genes by RT-PCR and western blot. As expected, based on the previous studies, we confirmed tyrosinase mRNA expression in both samples (Fig 6A). However, tyrosinase and GAPDH mRNA expression were slightly reduced in copper treatment samples compared to the non-copper controls, but this was not statistically significant (Fig 6B). Also, we detected DDC mRNA by the end-point RT-PCR (Fig 6A). On the other hand, tyrosine hydroxylase (TH) and dopamine beta-hydroxylase (DBH) were not detected by RT-PCR (S4 Fig). These mRNA expression results are consistent with previous RNA-seq data of ARPE19 (GSE88848) [44]. Furthermore, we confirmed tyrosinase protein expression by western blot (Figs 6C and 6D). Interestingly, copper supplementation increased tyrosinase protein expression. Thus, dopamine biosynthesis in ARPE19 by copper supplementation may be due to stabilization of tyrosinase and/or enhanced activity because tyrosinase converts tyrosine to L-DOPA.

## Discussion

We determine that copper supplementation efficiently enhanced pigmentation and induced dopamine biosynthesis in long-term ARPE19 culture. This is the first report showing the significance of copper to non-neuronal dopamine production in RPE cells, and it suggests that RPE-derived dopamine could potentially affect choroid homeostasis and myopia development.

In this study, we demonstrate a potential new pathway of dopamine production via tyrosinase in RPE cells stimulated by copper. Dopamine synthesis pathway in non-neuronal tissues has not been investigated well. We hope our results will trigger the study of dopamine production in non-neuronal tissues.

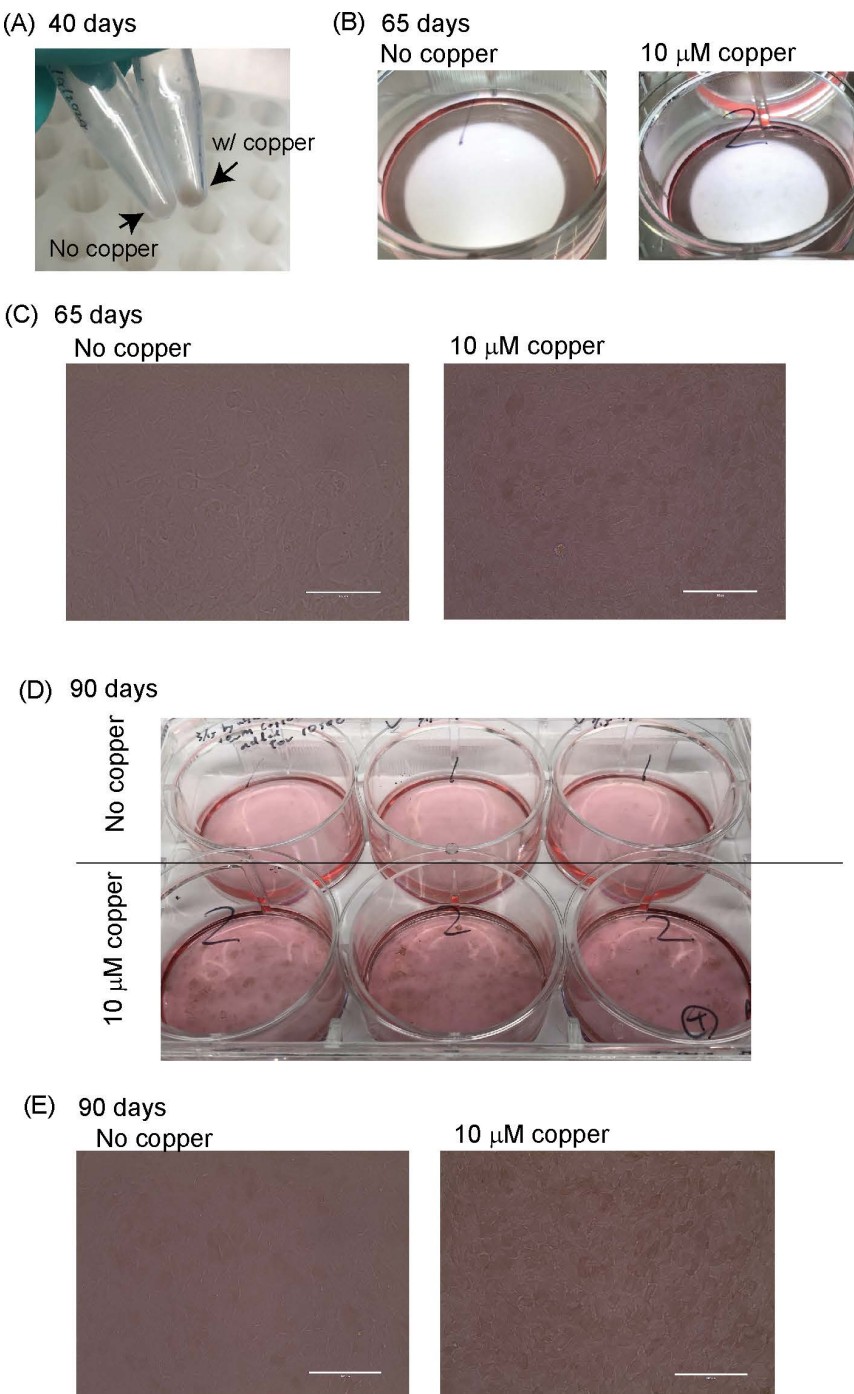

(A) 40 days

w/ copper

No copper

(B) 65 days
No copper          10 μM copper

(C) 65 days
No copper                    10 μM copper

(D) 90 days
No copper
10 μM copper

(E) 90 days
No copper          10 μM copper

**Fig 2. Copper (copper sulfate) enhances pigmentation in long-term ARPE19 culture.** (A) Cell pellet picture from 40-day ARPE19 culture. (B) 65-day cell culture picture and (C) images of bright field microscopy. (D) 90-day cell culture picture and (E) images bright field microscopy. Scale bar is 100 μm.

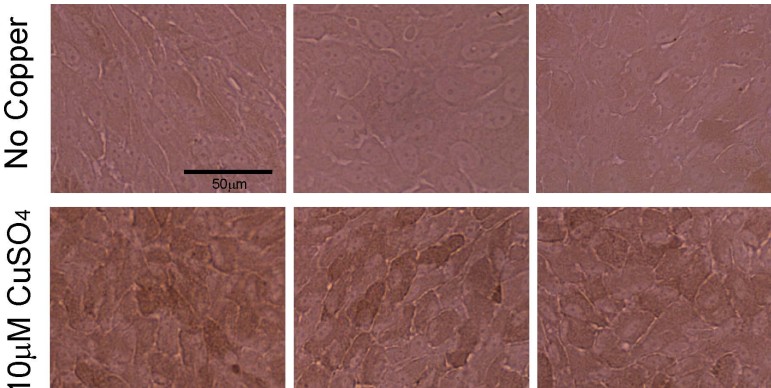

**Fig 3. Bright-field images of ARPE19 at high magnification.** Top: No copper, Bottom: 10 µM CuSO$_4$, 90 days culture. The scale bar is 50 µm.

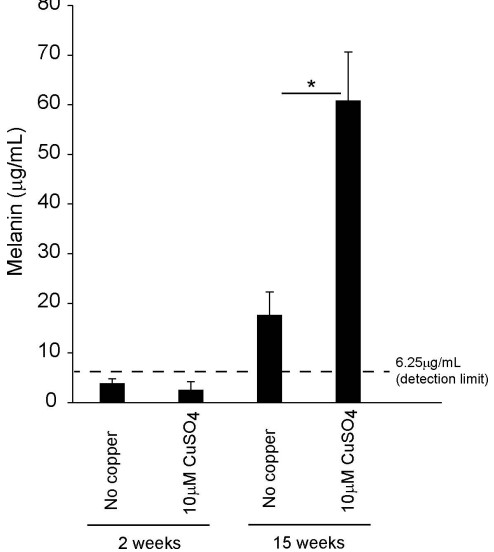

**Fig 4. Copper increases melanin levels in long-term ARPE19 culture.** Melanin concentration was measured in each cell lysate using OD500. Total protein concentration was adjusted at 2 mg/mL. N = 3. * indicates p < 0.01 by Student's t-test.

## Potential biological functions of RPE-derived dopamine

Dopamine's role in eye growth, choroidal thickness, and myopia development are areas of intense investigation. Our results indicate that RPE-derived dopamine may have biological roles in eye growth and homeostasis. In albinism, which is often caused by mutations of tyrosinase and tyrosinase-related protein, various ocular abnormalities, including myopia, decreased choroidal thickness, and decreased visual acuity were observed [47,48]. Although various gene mutations cause albinism, tyrosinase-dependent dopamine production may be associated with ocular phenotypes in albinism. In addition, retinal morphology and visual function in a murine model of albinism were rescued by L-DOPA administration without pigmentation [49], suggesting that tyrosinase-dependent L-DOPA and subsequent dopamine could be associated with ocular phenotypes in albinism.

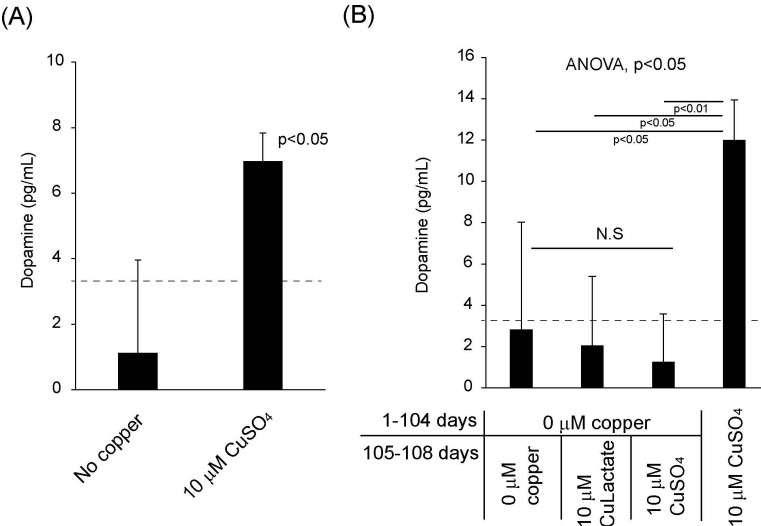

**Fig 5. Copper supplementation induced dopamine production in ARPE19 long-term culture.** (A) 90-day culture with 10 μM copper sulfate showed dopamine detected. N = 3. (B) After 104 days of culture, we added 10 μM copper lactate and 10 μM copper sulfate for 4 days. N = 3 (three individual samples). The error bar is standard deviation. The dotted lines indicate analytical sensitivity (3.3 pg/mL). ANOVA and Student's t-test were used.

## Copper stabilizes tyrosinase protein

Interestingly, in our experiment, copper supplements increased tyrosinase protein expression but not mRNA expression (Figs 6A and 6C). Tyrosinase requires multiple steps for maturation in the ER (endoplasmic reticulum) and Golgi, with strict quality controls. Although copper is not indispensable for tyrosinase processing/translocation, the holo-enzyme of tyrosinase showed higher stability than apo-enzyme [50]. Therefore, our result suggests that copper enhances tyrosinase protein stability. Also, dopamine production was not induced when copper was added to ARPE19 after a long-term culture without copper (Fig 5B). This may be due to inefficient copper transport into ARPE19 cells. Copper trafficking is highly regulated since an abundance of free copper is toxic due to its redox activity and potential to generate reactive oxygen species [51,52]. Ceruloplasmin, a major copper-carrying protein in the blood, plays a key role in systemic copper transport [53]. Although serum contains ceruloplasmin as a copper transporter [54], 1% FBS for the ARPE19 maturation may not have sufficient ceruloplasmin for efficient copper transport. Within cells, copper is delivered by transporters like CTR1 (also known as SLC31A) and chaperones such as ATOX1, while ATP7A/B and ceruloplasmin ensure proper distribution and export [55,56]. These genes may be downregulated under *in vitro* conditions. Alternative mechanisms of copper-mediated increase in tyrosinase protein levels could include translational activation or open reading frame operon regulation; these are less likely given the lack of apoptosis and the lack of multiple tyrosinase bands [57,58]. Thus, further investigation into copper transport may need to be done.

## Origin of DDC, exogenous or endogenous

DDC mRNA was detected in both copper-treated and non-treated ARPE19, but the expression level may be low because only end-point PCR detected it (Fig 6A). In addition to neurons, DDC is expressed in non-neuronal tissues [15,18,20,59], lymphocytes [17], and serum/plasma [60,61]. This suggests that serum in the culture medium may supply DDC. In human intact RPE, DDC expression was confirmed by RNA-seq (GSE159435) [62]. Therefore, in *in vivo* RPE/choroid, whether DDC is supplied endogenously and/or from the bloodstream (as the RPE is adjacent to the highly vascularized choroid) is unclear. Thus, further investigations are necessary to understand the precise mechanism of converting L-DOPA to dopamine.

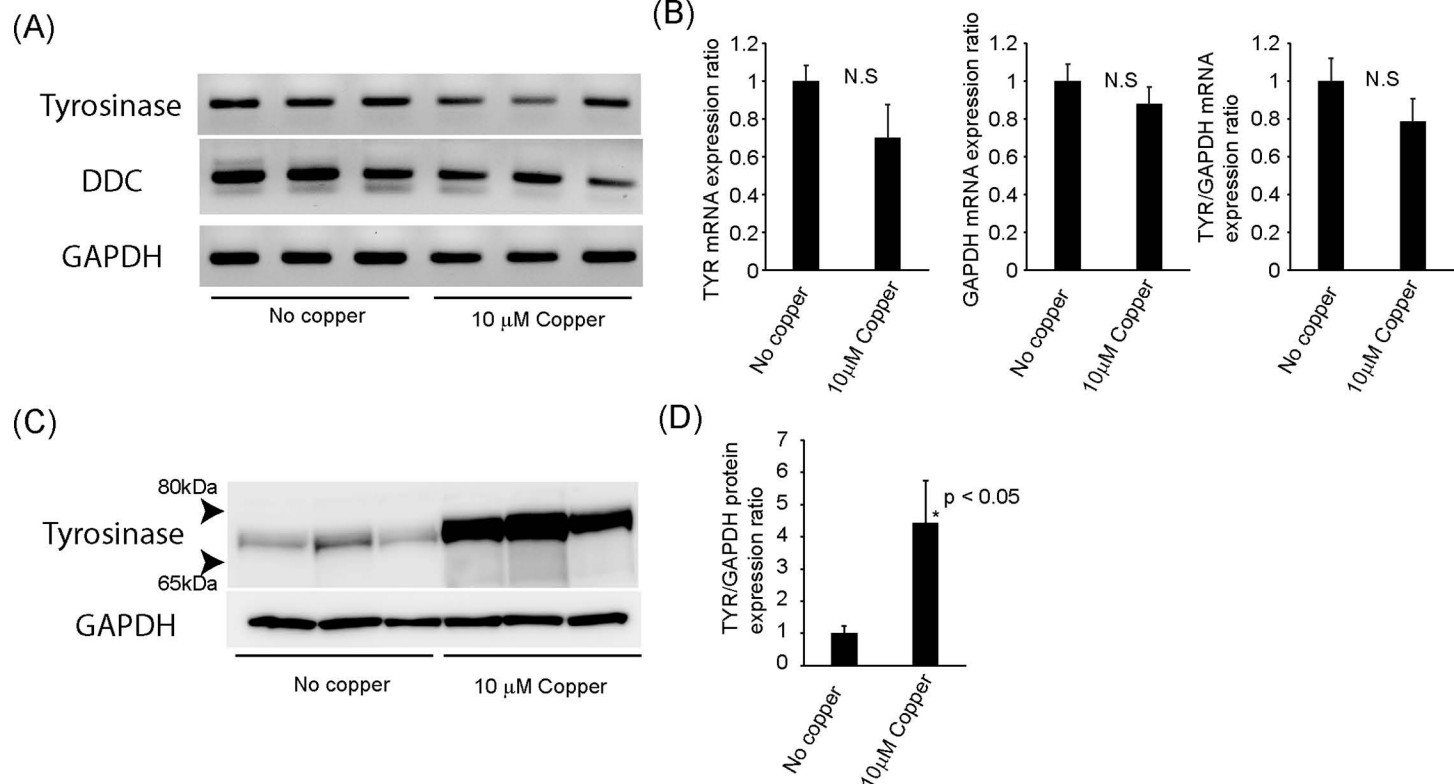

**Fig 6. Copper supplementation increases tyrosinase protein expression but not mRNA expression in long-term ARPE19 culture.** (A) RT-PCR results for tyrosinase, DDC, and GAPDH from 104-day culture. N = 3. As noted, DDC was amplified by the end-point PCR (40 cycles). (B) Densitometry of PCR bands of each gene. Student's t-test did not show a significant difference (N.S). (C) Western blot results for tyrosinase from 105-day culture. N = 3. (D) Densitometry of western blot bands of tyrosinase and GAPDH.

### Risk of tyrosinase inhibitors for peripheral dopamine production

Tyrosinase inhibitors have been extensively studied for cosmetic purposes [63]. However, our results indicate a potential risk of tyrosinase inhibitors to reduce dopamine in peripheral tissues. Tyrosinase has three different enzymatic reactions [40]. Among them, the reaction converting L-DOPA to dopaquinone (Fig 1) can be a target to inhibit melanin production without dopamine reduction. Also, the inhibition of this reaction could accumulate L-DOPA, which would result in an increase in dopamine.

### Limitations of the current study

We recognize several limitations of the current study. First, ARPE19 cells were used. Although ARPE19 is a useful RPE cell line, various RPE markers are low or lost compared to intact RPE and induced pluripotent stem cell (iPSC)-derived RPE [64,65]. Also, the RPE cell polarity was ignored [66,67]. Since our main subject was whether tyrosinase activation by copper can induce dopamine production, we believe that ARPE19 cells were sufficient. However, intact RPE or iPSC-derived RPE will be employed for further investigation. Second, only dopamine was analyzed by ELISA. Generally, dopamine is unstable and metabolized/processed into 3-Methoxytyramine (3-MT), 3,4-Dihydroxyphenylacetic acid (DOPAC), Homovanillic acid (HVA), Norepinephrine (noradrenaline) and Epinephrine (adrenaline) [68,69]. To understand dopamine production more precisely, we will assess these metabolites *in vivo* in future studies.

## Conclusion

In conclusion, we demonstrated that copper supplementation enhances pigmentation and induces dopamine production in long-term cultured ARPE19. Although we have not assessed its biological functions, RPE-derived dopamine could be involved in eye growth, choroidal homeostasis, and myopia development.

## Supporting information

**S1 Fig. CCK-8 assay for copper toxicity.** 1 day and 4 days with 10 µM copper sulfate did not show significant toxicity. * indicates $p < 0.05$ with Student's t-test.
(TIF)

**S2 Fig. Crystal violet assay for copper toxicity.** (A) Plate images after crystal violet staining. (B) After eluting the crystal violet, OD590 was measured. We found that copper treatment showed more cell growth rather than toxicity in our condition. N = 6. Student's t-test was used for average comparison.
(TIF)

**S3 Fig. Protein concentration in the lysis buffer after 105 days culture.** After 105 days culture, the cells were lysed with 250 µL RIPA buffer, and the protein concentration was measured by BCA assay. N = 3. * indicates $p < 0.05$ by Student's t test.
(TIF)

**S4 Fig. RT-PCR showed that ARPE19 did not express TH and DBH mRNA.** As a positive control, the same amount of human adrenal total RNA (Zyagen, San Diego, CA. Cat.no: HR-501) was used.
(TIF)

## Acknowledgments

The authors thank all members of the University of Oregon's Knight Campus for Accelerating Scientific Impact who helped with the experiments.

## Author contributions

**Conceptualization:** Hironori Uehara, Balamurali Ambati.

**Data curation:** Hironori Uehara.

**Formal analysis:** Hironori Uehara.

**Funding acquisition:** Hironori Uehara, Balamurali Ambati.

**Investigation:** Hironori Uehara, Baila Shakaib, Sangeetha Ravi Kumar.

**Methodology:** Hironori Uehara.

**Project administration:** Hironori Uehara.

**Supervision:** Hironori Uehara, Balamurali Ambati.

**Validation:** Hironori Uehara.

**Writing – original draft:** Hironori Uehara.

**Writing – review & editing:** Hironori Uehara, Bonnie Archer, Balamurali Ambati.

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
