## [Decision Letter · Decision Letter 0]

PONE-D-25-05806Copper supplementation enhances pigmentation and induces dopamine production in ARPE19PLOS ONE

Dear Dr. Uehara,

Thank you for submitting your manuscript to PLOS ONE. After careful consideration, we feel that it has merit but does not fully meet PLOS ONE’s publication criteria as it currently stands. Therefore, we invite you to submit a revised version of the manuscript that addresses the points raised during the review process.

The reviewers' major concerns pertain to the integrity of the cells and the quantitative measurement of pigmentation. Please provide additional evidence and clarification regarding the viability and structural integrity of the cells used in your experiments. Further, the methodology for measuring and analyzing pigmentation requires more rigorous quantitative approaches and appropriate statistical validation. Please thoroughly address these concerns along with all other reviewer queries in your revision.

We look forward to receiving your revised manuscript.

Kind regards,

Subbulakshmi Chidambaram, Ph.D

Academic Editor

PLOS ONE

Journal Requirements:

Reviewers' comments:

Reviewer's Responses to Questions

**Comments to the Author**

1. Is the manuscript technically sound, and do the data support the conclusions?

Reviewer #1: Yes

Reviewer #2: Yes

Reviewer #3: No

2. Has the statistical analysis been performed appropriately and rigorously? 

Reviewer #1: No

Reviewer #2: Yes

Reviewer #3: No

3. Have the authors made all data underlying the findings in their manuscript fully available?

Reviewer #1: Yes

Reviewer #2: Yes

Reviewer #3: No

4. Is the manuscript presented in an intelligible fashion and written in standard English?

Reviewer #1: Yes

Reviewer #2: Yes

Reviewer #3: Yes

5. Review Comments to the Author

Reviewer #1: In this article, the authors investigates the effect of copper supplementation on dopamine biosynthesis in ARPE19 cells, proposing a pathway in which tyrosinase mediates dopamine biosynthesis in RPE cells. The results are interesting, particularly in reporting copper-induced dopamine synthesis in non-neuronal cells. However, the manuscript requires further refinement in mechanistic exploration and experimental rigor. Below are specific comments:

1. The image quality of Figures 1C and E is insufficient to clearly discern changes in cellular pigmentation. Additionally, quantitative analysis of melanin should be included.

2. Quantitative analysis for Tyrosinase protein expression, showed in Figure 4C, should be included to clarify its role in dopamine synthesis.

3. Include cell viability assays (e.g., MTT or CCK-8) to rule out potential cytotoxicity from copper supplementation.

4. It is better to provide additional mechanistic evidence for tyrosinase protein stabilization, such as protein degradation assays (e.g., CHX chase) to confirm copper's role.

5. Provide more details on sample handling for dopamine ELISA, particularly measures to prevent dopamine degradation.

6. More background on the rationale for using ARPE19 cells as a model, especially in comparison to other RPE models (e.g., iPSC-derived RPE), would strengthen the introduction.

7. The discussion on "non-neuronal dopamine biosynthesis" could be expanded, particularly regarding existing research on dopamine sources in peripheral tissues.

8. Further explore the reasons for the lack of dopamine synthesis with short-term copper supplementation (Figure 3B), such as copper transport efficiency or intracellular copper dynamics.

9. Provide more literature support for the biological functions of RPE-derived dopamine, especially its mechanisms in choroidal thickness and myopia development.

10. Define abbreviations (e.g., DDC, L-DOPA) upon first use.

Reviewer #2: In this manuscript Uehara and coworkers test the hypothesis that tyrosinase-derived L-DOPA could be converted to dopamine and show that supplementation of ARPE19 medium with copper putatively increases pigmentation. The study tests an interesting hypothesis that could be of significant importance. The authors do show an increase in dopamine production using ELISA which is supportive of the proposed hypothesis, however the study as performed is incomplete for the following reasons. First the authors do little to quantify and characterize the putative increase in pigment induced by copper supplementation. They depend on qualitative visual observations and frankly the quality of figure 2 is less than compelling. Second, While they show an increase in dopamine expression by ELISA, they focus on tyrosinase expression and ignore expression of aromatic L-amino acid decarboxylase (AADC), the enzyme that catalyzes the conversion of L-dopa to dopamine. The authors could significantly improve the manuscript by addition of the following:

1. Melanin pigment should be quantitatively assessed. One approach might be to use a spectrophotometric assay like that of Hu (PMID: 18435617).

2. Figure 2 is not terribly compelling as a visual representation of pigmentation. Better quality photos are required and should include photomicrographs at a cellular level showing pigment granules.

3. The authors should examine expression of AADC. This should include assay for AADC mRNA and/or protein as well as assay of AADC activity in cell lysates both with and without Copper supplementation.

Reviewer #3: The authors have conducted a study investigating enhanced pigmentation and dopamine production after copper supplementation in ARPE19 cells. The outline of the study is well understood, however there are some shortcomings that lead me to suggest that the study not be accepted.

- Statistical accounting is missing overall. This is a serious flaw.

- ARPE19 cells have high turnover and usually used for many cycles. They can easily undergo a "drift" over time, which can express itself genetically, morphologically, in growth patterns, etc. The authors need to at least include check points to control for the drift.

- Pigmentation has been assessed visually, but an objective measurement method needs to be introduced.

6. PLOS authors have the option to publish the peer review history of their article (what does this mean? ). If published, this will include your full peer review and any attached files.

**Do you want your identity to be public for this peer review?** For information about this choice, including consent withdrawal, please see our Privacy Policy .

Reviewer #1: No

Reviewer #2: No

Reviewer #3: No

---

## [Author Response · Author response to Decision Letter 1]

29 Apr 2025

Reviewer #1: In this article, the authors investigates the effect of copper supplementation on dopamine biosynthesis in ARPE19 cells, proposing a pathway in which Tyrosinase mediates dopamine biosynthesis in RPE cells. The results are interesting, particularly in reporting copper-induced dopamine synthesis in non-neuronal cells. However, the manuscript requires further refinement in mechanistic exploration and experimental rigor. Below are specific comments:

1. The image quality of Figures 1C and E is insufficient to clearly discern changes in cellular pigmentation. Additionally, quantitative analysis of melanin should be included.

We added higher magnification images (Supplemental Figure 4) and melanin quantification (Supplemental Figure 5).

2. Quantitative analysis for Tyrosinase protein expression, showed in Figure 4C, should be included to clarify its role in dopamine synthesis.

We quantified the tyrosinase band (Figure 4D). Also, we clarified tyrosinase activity (tyrosine to L-DOPA) in the result section.

3. Include cell viability assays (e.g., MTT or CCK-8) to rule out potential cytotoxicity from copper supplementation.

We conducted a CCK-8 assay for a short-term culture (1-4 days) and crystal violet for a mid-term culture (1-14 days). Since our ARPE19 culture for maturation would require over 90-100 days, the protein concentrations of the 105-day culture were evaluated for potential copper toxicity. In our conditions, we did not find significant toxicity; copper supplements (10uM) indicate that cell survival increased slightly. However, this result does not guarantee the increase of cell survival since copper also impacts the extracellular matrix (e.g., collagen cross-linking enzymes such as lysyl oxidase). Therefore, we concluded, "Copper did not show significant cell toxicity in our condition." These results are shown in Supplemental Figures 1-3.

4. It is better to provide additional mechanistic evidence for tyrosinase protein stabilization, such as protein degradation assays (e.g., CHX chase) to confirm copper's role.

We agree with the reviewer's point. However, there are several challenges in addressing this:

A) The long-term culture is necessary. We also recognize that this resulted in more deviations in tyrosinase expression between the samples.

B) Non-copper culture medium could contain residual copper from 1%FBS. Therefore, tyrosinase in non-copper-treated cells could bind copper from the medium.

C) Since copper is an indispensable element for cell survival, such as cytochrome c oxidase in mitochondria, the complete chelation of copper may cause malfunction of the cells.

D) Most copper binds to serum proteins such as albumin and ceruloplasmin. Removing the copper from the protein may require a special type of dialysis, such as albumin dialysis.

Therefore, instead of CHX experiments, the discussion about tyrosinase stability was expanded as below.

" Tyrosinase requires multiple steps for maturation in the ER (endoplasmic reticulum) and Golgi, with strict quality controls. Although copper is not indispensable for tyrosinase processing/translocation, the holo-enzyme of tyrosinase showed higher stability than apo-enzyme

5. Provide more details on sample handling for dopamine ELISA, particularly measures to prevent dopamine degradation.

About the sample handling, we added metabisulfite (final conc. 4mM) and EDTA (final conc. 1mM). Also, the samples were kept on ice until the experiment, and we started the ELISA within 30 minutes after harvesting the culture medium. We added these points to the method.

This ELISA kit contains unique methods to prevent dopamine: cis-diol affinity extraction, acylation, and enzyme conversion. This process prevents dopamine degradation.

We also asked the manufacturer for references regarding dopamine degradation; unfortunately, they do not have a paper to show the above, but they agreed with the above statement. Also, they said, "the dopamine would be most suspectable to degradation (pH> 7,5). However, the buffer added for extraction contains EDTA. Also important is that the standards undergo the same procedure, and if degradation occurs, this will also be the case for the standards."

6. More background on the rationale for using ARPE19 cells as a model, especially in comparison to other RPE models (e.g., iPSC-derived RPE), would strengthen the introduction.

Yes, we added more background about ARPE19 and its culture cost are much more affordable compared to the iPSC-derived RPE. Also, the provider of the iPSC-derived RPE does not disclose the composition of their culture medium. Application of the current culture condition to iPSC-derived RPE may cause unexpected results. Therefore, we used APRE19 in this study (copper effect on tyrosinase and dopamine). However, we will use iPSC-derived RPE for future experiments.

7. The discussion on "non-neuronal dopamine biosynthesis" could be expanded, particularly regarding existing research on dopamine sources in peripheral tissues.

In the introduction, we expanded the discussion of non-neuronal dopamine biosynthesis.

8. Further explore the reasons for the lack of dopamine synthesis with short-term copper supplementation (Figure 3B), such as copper transport efficiency or intracellular copper dynamics.

We expanded the discussion about copper transport.

9. Provide more literature support for the biological functions of RPE-derived dopamine, especially its mechanisms in choroidal thickness and myopia development.

We added additional references. However, most literature studied retinal dopamine (neuronal retina) or ocular dopamine (whole eye). Therefore., in the introduction, we discussed dopamine's general biological function, not specifically that of of RPE-derived dopamine.

10. Define abbreviations (e.g., DDC, L-DOPA) upon first use.

We apologize for the earlier oversight. We have now defined the abbreviations in the first appearance.

Reviewer #2: In this manuscript Uehara and coworkers test the hypothesis that tyrosinase-derived L-DOPA could be converted to dopamine and show that supplementation of ARPE19 medium with copper putatively increases pigmentation. The study tests an interesting hypothesis that could be of significant importance. The authors do show an increase in dopamine production using ELISA which is supportive of the proposed hypothesis, however the study as performed is incomplete for the following reasons. First the authors do little to quantify and characterize the putative increase in pigment induced by copper supplementation. They depend on qualitative visual observations and frankly the quality of figure 2 is less than compelling. Second, While they show an increase in dopamine expression by ELISA, they focus on tyrosinase expression and ignore expression of aromatic L-amino acid decarboxylase (AADC), the enzyme that catalyzes the conversion of L-dopa to dopamine. The authors could significantly improve the manuscript by addition of the following:

1. Melanin pigment should be quantitatively assessed. One approach might be to use a spectrophotometric assay like that of Hu (PMID: 18435617).

We added the melanin quantification result as Supplemental Figure 5.

2. Figure 2 is not terribly compelling as a visual representation of pigmentation. Better quality photos are required and should include photomicrographs at a cellular level showing pigment granules.

We added high-magnification images as Supplemental Figure 4.

3. The authors should examine expression of AADC. This should include assay for AADC mRNA and/or protein as well as assay of AADC activity in cell lysates both with and without Copper supplementation.

We examined DDC mRNA expression (Figure 4) as the equivalent of AADC expression. AADC is a more common name than DDC, but we followed HGNC (HUGO Gene Nomenclature Committee), which uses DDC as the approved name. To reduce the confusion, we stated this more clearly at the first appearance of DDC.

Reviewer #3: The authors have conducted a study investigating enhanced pigmentation and dopamine production after copper supplementation in ARPE19 cells. The outline of the study is well understood, however there are some shortcomings that lead me to suggest that the study not be accepted.

1. Statistical accounting is missing overall. This is a serious flaw.

We apologize for the earlier oversight, and we have now added a statistical section to the methods.

2. ARPE19 cells have high turnover and usually used for many cycles. They can easily undergo a "drift" over time, which can express itself genetically, morphologically, in growth patterns, etc. The authors need to at least include check points to control for the drift.

Thank you for your valuable feedback regarding genetic drift in our cell culture experiments. We agree that genetic drift can significantly impact genotype, phenotype, and overall cell functionality. Subculturing and increasing passage numbers are recognized as primary contributors to this drift. To minimize such effects, we prepared cryo-stocks of ARPE19 cells in passage 2 upon receipt from ATCC and used them for experiments in passages 4–5. Also, the cells were allowed to mature in 1% FBS without subculture. This approach should mitigate genetic drift, and we believe its impact in our experiments was minimal to moderate.

While cryopreservation is a common strategy to limit genetic drift and loss of cell characteristics, we were unable to perform confirmatory genomic analyses due to limited resources. We will consider including genetic drift assessments in future studies. We appreciate your understanding and the opportunity to address this point.

3. Pigmentation has been assessed visually, but an objective measurement method needs to be introduced.

We quantified the melanin in the cell lysate and showed it in Supplemental Figure 5.

---

## [Editor Report · Decision Letter 1]

PONE-D-25-05806R1Copper supplementation enhances pigmentation and induces dopamine production in ARPE19PLOS ONE

Dear Dr. Uehara,

Thank you for submitting your manuscript to PLOS ONE. After careful consideration, we feel that it has merit but does not fully meet PLOS ONE’s publication criteria as it currently stands. Therefore, we invite you to submit a revised version of the manuscript that addresses the points raised during the review process.

The changes made in response to the previous comments are satisfactory. However, we noted that some supplementary figures, especially, Suppl Fig.4 and 5 are important for the proper interpretation of your findings. Therefore, to ensure clarity and full accessibility of the key data to the readers, I kindly request you to place those images in the main manuscript and update the figure numbering accordingly.

We look forward to receiving your revised manuscript.

Kind regards,

Subbulakshmi Chidambaram, Ph.D

Academic Editor

PLOS ONE

Journal Requirements:

Additional Editor Comments :

The changes made in response to the previous comments are satisfactory. However, we noted that some supplementary figures, especially, Suppl Fig.4 and 5 are important for the proper interpretation of your findings. Therefore, to ensure clarity and full accessibility of the key data to the readers, I kindly request you to place those images in the main manuscript and update the figure numbering accordingly.

---

## [Author Response · Author response to Decision Letter 2]

6 Jun 2025

We appreciate the opportunity to submit the revised version of our manuscript. In response to the latest comment, we have revised the manuscript as follows:

1) S4 Fig and S5 Fig have been moved to Fig 3 and Fig 4, respectively.

2) Figure numbers and legends have been adjusted accordingly throughout the manuscript.

Please let us know if any further modifications are required. Thank you again for your time and consideration.

---

## [Editor Report · Decision Letter 2]

Copper supplementation enhances pigmentation and induces dopamine production in ARPE19

PONE-D-25-05806R2

Dear Dr. Uehara,

We’re pleased to inform you that your manuscript has been judged scientifically suitable for publication and will be formally accepted for publication once it meets all outstanding technical requirements.

Kind regards,

Subbulakshmi Chidambaram, Ph.D

Academic Editor

PLOS ONE
---

## [Editor Report · Acceptance letter]

PONE-D-25-05806R2

PLOS ONE

Dear Dr. Uehara,

I'm pleased to inform you that your manuscript has been deemed suitable for publication in PLOS ONE. Congratulations! Your manuscript is now being handed over to our production team.

Kind regards,

on behalf of

Dr. Subbulakshmi Chidambaram

Academic Editor

PLOS ONE